

# The root enrichment of bacteria is consistent across different stress-resistant plant species

Feng Huang[1], Congyi Zhu[2], Minli Huang[3], Xiaobing Song[1] and Aitian Peng[1]

[1] Institute of Plant Protection, Guangdong Academy of Agricultural Sciences, Key Laboratory of Green Prevention and Control on Fruits and Vegetables in South China Ministry of Agriculture and Rural Affairs, Guangdong Provincial Key Laboratory of High Technology for Plant Protection, Guangzhou, Guangdong, China
[2] Key Laboratory of South Subtropical Fruit Biology and Genetic Resource Utilization (MOA) & Guangdong Province Key Laboratory of Tropical and Subtropical Fruit Tree Research, Institute of Fruit Tree Research, Guangdong Academy of Agricultural Sciences, Guangzhou, Guangdong, China
[3] Lichuan Bureau of Natural Resources, Fuzhou, Jiangxi, China

## ABSTRACT

Bacteria, inhabiting around and in plant roots, confer many beneficial traits to promote plant growth and health. The secretion of root exudates modulates the nutritional state of the rhizosphere and root area, further selecting specific bacteria taxa and shaping the bacteria communities. Many studies of the rhizosphere effects have demonstrated that selection by the plant rhizosphere consistently enriches a set of bacteria taxa, and this is conserved across different plant species. Root selection effects are considered to be stronger than the rhizosphere selection effects, yet studies are limited. Here, we focus on the root selection effects across a group of 11 stress-resistant plant species. We found that the root selection consistently reduced the alpha diversity (represented by total number of observed species, Shannon's diversity, and phylogenetic diversity) and altered the structure and composition of bacteria communities. Furthermore, root selection tended to enrich for clusters of bacteria genera including *Pantoea*, *Akkermansia*, *Blautia*, *Acinetobacter*, *Burkholderia-Paraburkholderia*, *Novosphingobium*, *Massilia*, *Pseudomonas*, *Chryseobacterium*, and *Stenotrophomonas*. Our study offers some basic knowledge for understanding the microbial ecology of the plant root, and suggests that several bacteria genera are of interest for future studies.

## INTRODUCTION

The plant root and surrounding soil area represents a hotspot for interactions between plants and microbes (*Mitter, De Freitas & Germida, 2017*; *Zhalnina et al., 2018*). Two distinct compartments, the rhizosphere and the root or endosphere, are separated based on their physical proximity to the plant and the level of host's influence on the microbial communities assembled from the pool of soil (*Fitzpatrick et al., 2018*; *Yurgel et al., 2018*). The rhizosphere compartment is inhabited by microbes that assemble sourrounding the roots, whereas the root compartment is inhabited by the microbes assembled within roots (*Fitzpatrick et al., 2018*; *Yurgel et al., 2018*). Both compartments are inhabited by a highly

Corresponding authors
Xiaobing Song, 527303019@qq.com
Aitian Peng, pengait@163.com

diverse species of microbes, especially bacteria, that provide essential ecosystem services such as nutrient cycling, soil structuring, stress alleviation, and resource use efficiency to the plants (*Saleem et al., 2018*).

Even though the rhizosphere and root areas are diverse and rich in organic carbon and other nutrients, only a small set of microbes can thrive in these areas (*Huang et al., 2014*; *Cordovez et al., 2019*). For example, when studying the root microbiome of different plants across lycopods, ferns, gymnosperms, and angiosperms, bacteria genera *Bradyrhizobium*, *Rhizobium*, and *Burkholderia* were consistently found as core members (*Yeoh et al., 2017*). Similarly in an agricultural system, the root interior of four crops: canola, wheat, field pea, and lentil were dominated by bacteria genera including *Pseudomonas*, *Stenotrophomonas*, *Acinetobacter*, *Arthrobacter*, *Rhizobium*, *Streptomyces*, *Variovorax*, and *Xanthomonas* (*Cordero, De Freitas & Germida, 2020*). The microbe selection effects of the rhizosphere and the root system imply that their microbial community is structured by a complex series of interactions and feedbacks between plant roots, microbes, and the physical and chemical environment of the soil (*Zhalnina et al., 2018*; *Cordovez et al., 2019*). In the rhizosphere, plant exudates include many small signaling molecules (*Hassan & Mathesius, 2012*; *Huang et al., 2014*), polymers (*Beauregard et al., 2013*), antimicrobials (*Huang et al., 2014*), and plant hormones (*Lebeis et al., 2015*) which have been substantially demonstrated to function in the selection of different microbal taxa. The root then selects microbes from the rhizosphere community, though the root selection patterns and enrichment of these microbial taxa have been less studied.

This study focused on the root selection of soil bacteria across a group of 11 stress-resistant plant species, all of which are wild populations that are commonly present in Hainan Province, China. Based on their ability to resist harsh environmental conditions and barren soils, these plant species have been recommended as pioneer plants to restore destroyed and degraded ecosystems (*Tong et al., 2013*; *Liu et al., 2015*; *Ren et al., 2017*). For example, *Wang et al. (2019)* demonstrated that the growth of different combinations of these plants significantly increased soil water content, microbial biomass, carbon, nitrogen, and decreased soil pH. Of these, *Calophyllum inophyllum* (CI) and *Guettarda speciosa* (GS) showed strong levels of stress resistance (*Zhang et al., 2019*; *Li et al., 2021*). In addition, the plant level of stress-resistance is highly associated with root bacteria communities (*Ali, Charles & Glick, 2014*; *Sandhya et al., 2017*). However, with the large abundance of root-inhabiting bacteria, the usage of culture-dependent methods to study this process and to screen potential beneficial bacteria is arduous. Our study aimed to characterize root selection of bacteria from the rhizosphere soils across different plant species, and elucidate whether the root selection reduces the alpha diversity (represented by total number of observed species, Shannon's diversity, and phylogenetic diversity) and alters the structure and composition of bacteria communities. Furthermore, the roots consistently enriches a similar cluster of bacteria genera which may harbor key functional genes for their survival and plant-beneficial traits within root. Finally, our study suggests several bacteria species that may share a close relationship to plant and worth for experimental tests.

## MATERIALS AND METHODS

### Plant growth and sampling

Eleven plant species, including *Casuarina equisetifolia* (CE), *Calophyllum inophyllum* (CI), *Fagraea ceilanica* (FC), *Guettarda speciosa* (GS), *Heritiera littoralis* (HL), *Hernandia sonora* (HS), *Melaleuca bracteata* (MB), *Pongamia pinnata* (PP), *Portulaca pilosa* (PPI), *Ruellia brittoniana* (RB), and *Scaevola sericea* (SS), were selected in this study based on their ability to thrive under harsh environments, such as high salinity and alkalinity soils and sites with high temperatures and light, and under drought (*Tong et al., 2013*; *Liu et al., 2015*; *Ren et al., 2017*). Plant seeds were sown in pots (24 × 26 cm, diameter × height) filled with field soils (pH = 8.0∼9.0, mean soil organic carbon 0.4%, total nitrogen 0.05%, total phosphorus 0.03%) collected near a greenhouse of Lingye Gardening Limited Compary (Wenchang, China). For each plant species, four seeds were sown in each pot of a total of six pots. After germination, only one seedling was kept growing in each pot, and the remaining plants were grown in the same greenhouse from February to August, 2017. All plant and soil samples were harvested in August 14, 2017. The soils collected from the field, incubated in the pots for the same period to the plant growth, were designated as bulk soil, and the soils, adhered to plant roots, were mechanically brushed and kept as rhizosphere soil. Fine roots, with diameter smaller than two mm, were washed in 1 × TE buffer (Tris-EDTA buffer: 10 mM Tris, 1 mM EDTA, pH = 8.0. Added with 0.1% Triton X-100) for 30s, 75% ethanol for 15s, 2% bleach for 15s, and rinsed three times with sterile water (*Agler et al., 2016*). The soils and sterile roots were stored under −80 °C overnight before DNA extraction.

### DNA extraction and amplicon sequencing

Microbial genomic DNA was extracted from 250 mg soil or 100 mg root with the Powersoil® Kit (MoBio Laboratories Inc., Carlsbad, CA, USA). The primer pair 515F/806R was used to amplify partial sequence of the 16S V4-V5 rRNA region of the bacteria community (*Caporaso et al., 2011*). PCR reactions were carried out using Phusion High-Fidelity PCR Master Mix (New England Biolabs Inc., Ipswich, MA, USA). The PCR cycle was set with an initial 94 °C for 5 min, followed by 35 cycles of 94 °C for 45s, 56 °C for 30s, and 72 °C for 30s, a final extension of 72 °C for 10 min. The harvested DNA was verified on a 2% agarose electrophoresis gel and prepared for sequencing using the TruSeq® DNA PCR-Free Library Preparation Kit (Illumina, San Diego, CA, USA). Amplicon sequencing was carried out using the Illumina HiSeq2500 platform (Illumina, San Diego, CA, USA).

### Bioinformatics analysis

Paired-end reads were merged by FLASH v1.2.7 (*Magoč & Salzberg, 2011*) and spliced raw tags were quality filtered by QIIME v1.7.0 (http://qiime.org/) to obtain high-quality clean tags (*Caporaso et al., 2010*). Chimeric sequences were removed by following UCHIME algorithm (*Edgar et al., 2011*). Then, the effective tags were assigned to Operational Taxonomic Units (OTUs) by Uparse v7.0.1001 (http://drive5.com/uparse/) at 97% similarity level (*Edgar, 2013*). All singletons were deleted, the remaining sequences of each sample were normalized to the sample with the least number of sequences (32,416). The rarefaction curve showed
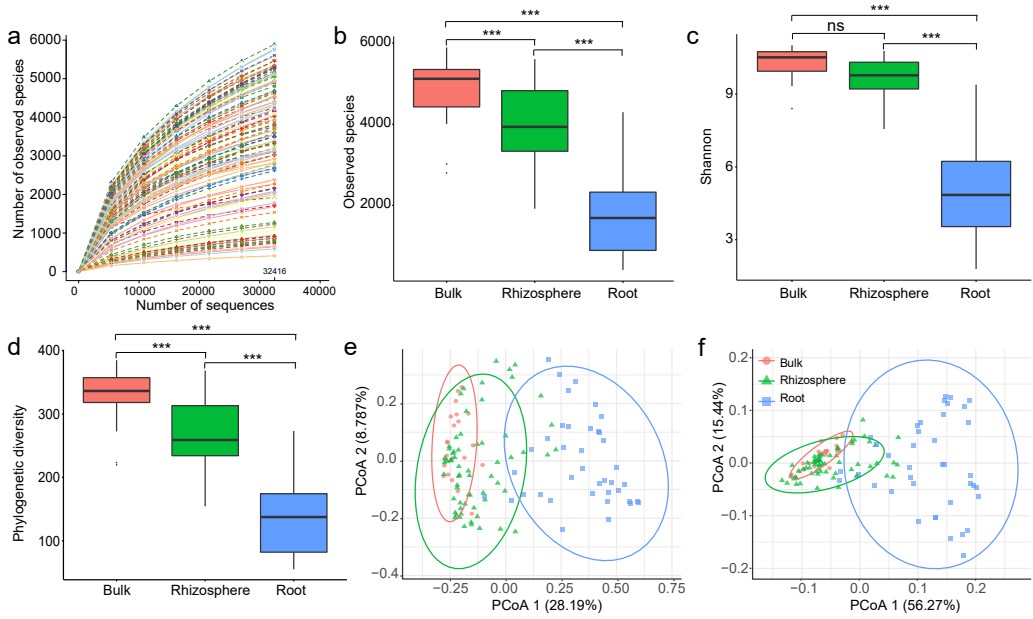

**Figure 1 The overall comparison of the bacteria communities among the bulk soil, the rhizosphere soil, and the root.** (A) The rarefaction curves of sequenced samples rarefied at 32,416 sequences. The number of observed species (B), Shannon's diversity (C), and phylogenetic diversity (D); the principal component analysis of bacteria communities (E) and predicted genes (F) among the samples of the bulk soil, the rhizosphere soil, and the root. The $p$ values from a Tukey comparison test are marked with "ns" for $p \geq 0.05$, "*" for $p < 0.05$, "**" for $p < 0.005$, "***" for $p < 0.0005$.

the number of OTUs/observed species identified in randomly picked sequences, the curve of each sample reached to plateau, implying that deeper sequencing might not result in a significant increase of the number of observed species (Fig. 1A). The taxonomic identity was assigned to bacteria OTUs after searching against the SILVA138 database (*Yilmaz et al., 2014*).

## Statistical analysis

The alpha diversity indices (Observed species, Chao1, Shannon, Simpson, ACE, Good's coverage, and Phylogenetic diversity), summarizing the distribution of species abundances and diversities in a given sample (*Thukral, 2017*), were calculated in QIIME v1.9.1 and visualized by using R software v4.0.2 (https://www.r-project.org/). For example, the Shannon's diversity index was calculated with Sum ($p_i\log(p_i)$), the phylogenetic diversity value was calculated as the sum of branch lengthes of all species in a given sample (*Faith, 1992*; *Thukral, 2017*). Differences of bacteria community structure among the bulk soils, rhizosphere soils, and roots were assessed by performing PERMANOVA with 999 permutations in the *vegan* package in R (*R Core Team, 2020*). A principal components analysis (PCoA) based on bray-curtis dissimilarities was performed using the cmdscale function within *vegan* package. Function prediction of each OTU was performed by Tax4Fun (*Aßhauer et al., 2015*). The estimation of fold change of bacteria taxa and predicted functions was performed in DESeq2 (*Love, Huber & Anders, 2014*). Equality

of variances (bartlett.test) and condition of normality (shapiro.test) were tested in R software, ANOVAs followed by Tukey *post-hoc* pairwise test was used for comparisons among the bulk soils, rhizosphere soils, and root samples, two-tailed $t$-test was used for comparisons between the rhizosphere soils and the root samples. $P$ values were adjusted using the Benjamini–Hochberg method, and $p$ values smaller than 0.05 were accepted with significance.

# RESULTS

## Overall community differences

A total of 125 samples, including 22 of bulk soils, 64 of rhizosphere soils, and 39 of plant roots, were successfully sequenced, resulting in 8,946,878 16S rRNA sequences (mean 71,575 ± standard deviation 11,164 per sample). All sequences were clustered into 24,202 Operational Taxonomic Units (OTUs, on average 3450 ± 1513 per sample) by Uparse v7.0.1001 (http://drive5.com/uparse/) at 97% similarity level. Compared to bulk and rhizosphere soil samples, root samples showed significantly reduced numbers of observed species (4890 ± 835, 3998 ± 923 to 1737 ± 978, $p < 0.01$, Fig. 1B), Shannon's diversity (10.3 ± 0.6, 9.7 ± 0.8 to 5.1 ± 1.8, $p < 0.01$, Fig. 1C), and phylogenetic diversity (329.2 ± 45.6, 269.1 ± 50.6 to 137.9 ± 57.2, $p < 0.01$, Fig. 1D). When examing beta diversity, root samples were also significantly differentiated from soil samples in composition and structure (PCoA 1 = 28.2%, $p < 0.01$, Fig. 1E), and also in predicted functions (PCoA 1 = 56.3%, $p < 0.01$, Fig. 1F).

## Comparison of rhizosphere and root communities by plant species

In overall analysis, root bacteria communities were composed of different OTUs and overall less diversity compared to the rhizosphere bacteria communities. This phenomenon was consistent across all tested plant species. For each plant species, the observed species, Shannon's diversity, and phylogenetic diversity were typically lower in root samples compared to the rhizosphere samples (Fig. 2). Specifically, there was a significant reduction in alpha diversity estimates, observed species (Fig. 2A), Shannon's diversity (Fig. 2B), and phylogenetic diversity (Fig. 2C) in seven (CE, CI, FC, GS, HS, MB, and PPI) out of 11 plant species (63.6%) when comparing the rhizosphere soil communities with the root communities. Similarly, clear separation in bacteria structure was observed between the rhizosphere soil communities and the root communities in ten plant species (90.9%, except PP) using just one principal component (PCoA 1, Fig. 3A). Further separation was observed in four plant species (36.4%, CE, GS, HL, PPI) with a second principal component (PCoA 2, Fig. 3B). Root selection and differentiation of the rhizosphere and root bacteria communities was most significant in plant species CE, CI, and GS, whereas no differences were observed in the bacteria communities of PP.

## Taxa enrichment and depletion in root compared to rhizosphere soil

To further characterize the differences in bacteria communities between the roots and the rhizosphere soils, the relative abundances of the top 50 bacteria genera and predicted genes were compared. The log fold change of bacteria genera and predicted genes in root, either

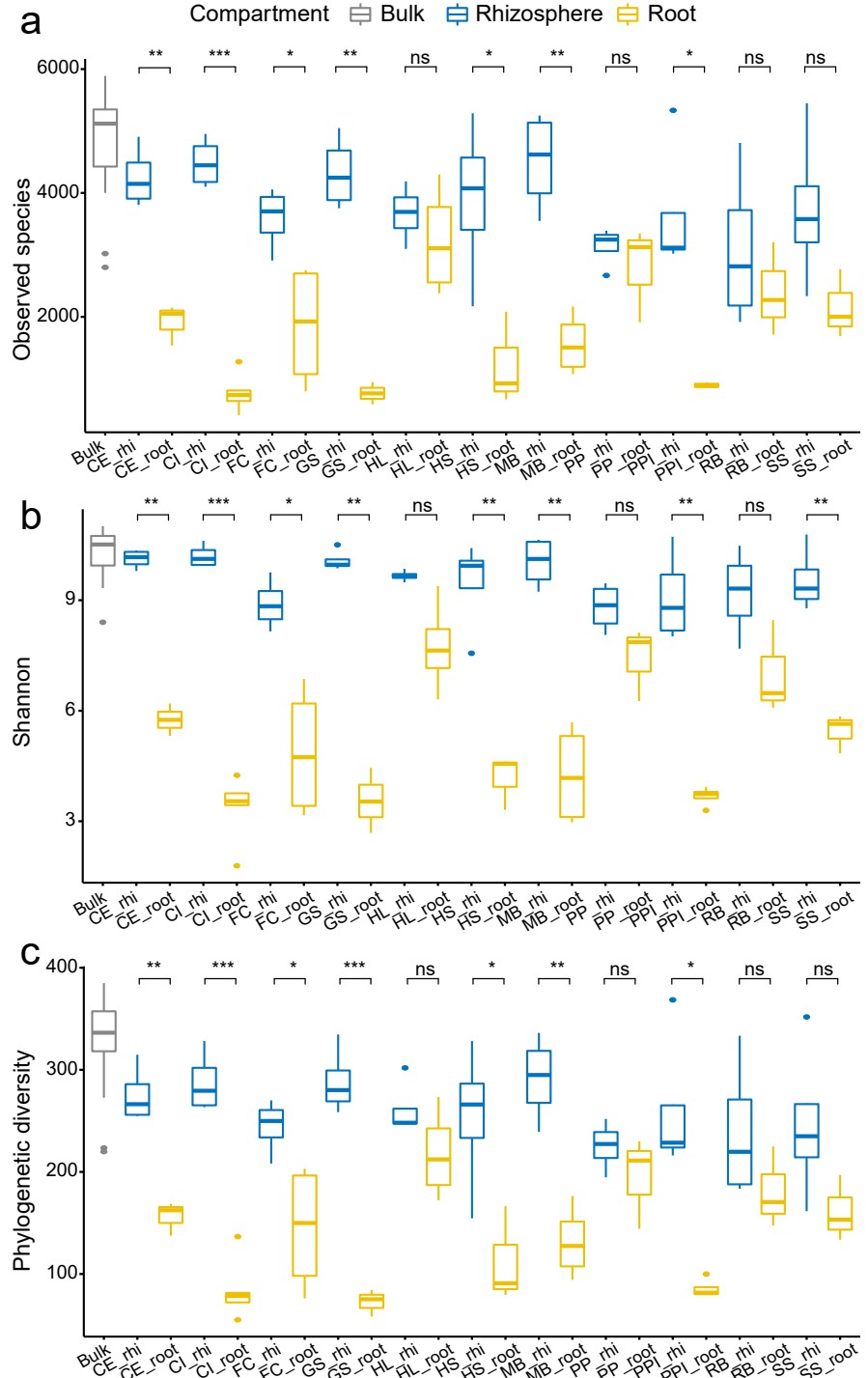

**Figure 2** **The comparison of the number of observed species (A) Shannon's diversity (B) and phylogenetic diversity (C) among the bulk soil, the rhizosphere soil, and the root by plant species.** The *p* values of the *t*-tests are marked with "ns" for $p \geq 0.05$, "*" for $p < 0.05$, "**" for $p < 0.005$, "***" for $p < 0.0005$. The abbreviations of the plant species: CE, *Casuarina equisetifolia*; CI, *Calophyllum inophyllum*; FC, *Fagraea ceilanica*; GS, *Guettarda speciosa*; HL, *Heritiera littoralis*; HS, *Hernandia sonora*; MB, *Melaleuca bracteata*; PP, *Pongamia pinnata*; PPI, *Portulaca pilosa*; RB, *Ruellia brittoniana*; SS, *Scaevola sericea*.

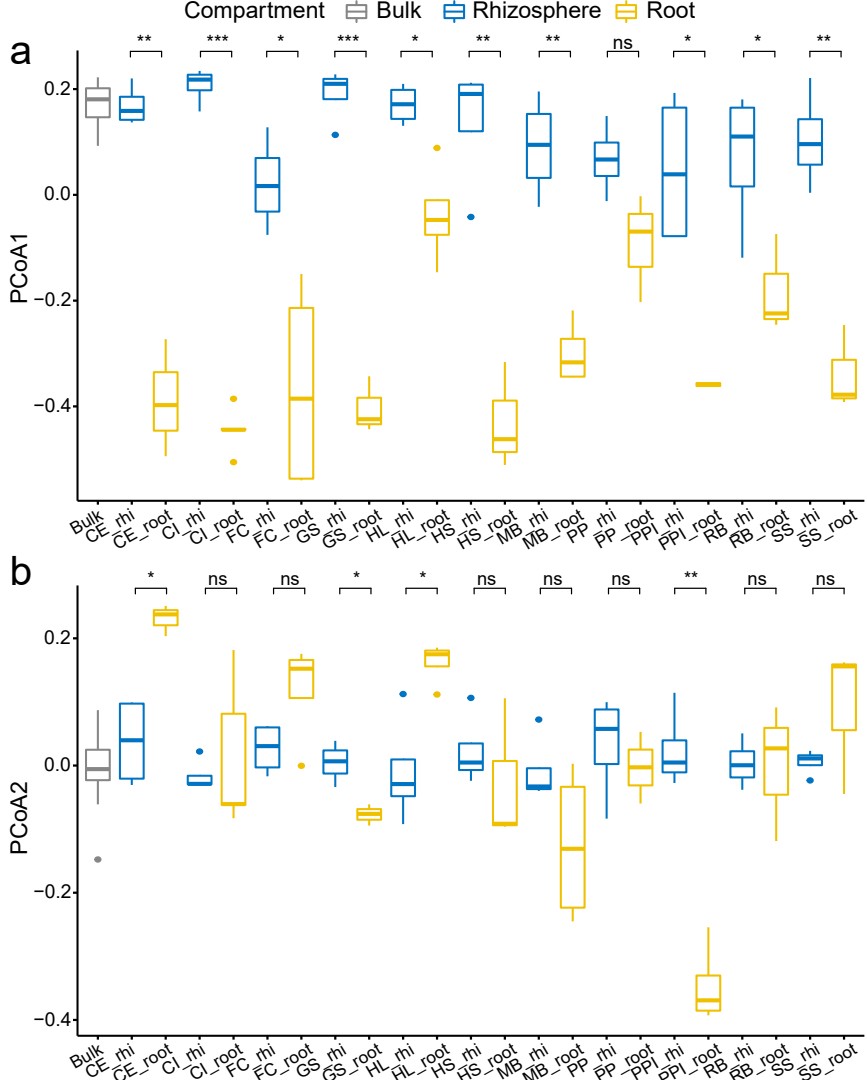

**Figure 3** **The comparison of the first two principle components PCoA 1 (A) and PCoA 2 (B) among the bulk soil, the rhizosphere soil, and the root by plant species.** The $p$ values of the $t$-tests are marked with "ns" for $p \geq 0.05$, "*" for $p < 0.05$, "**" for $p < 0.005$, "***" for $p < 0.0005$. The abbreviations of the plant species are listed in Fig. 2.

enriched or decreased, were plotted by plant species in heatmaps (Fig. 4). From the plot, a clade of bacteria including *Candidatus* Solibacter, *Pseudolabrys*, *Variibacter*, H16 (of the family *Desulfurellaceae*), *Bryobacter*, *Rhizomicrobium*, and *Haliangium* decreased in the roots of multiple plant species (Fig. 4A). On the contrary, *Pantoea*, *Akkermansia*, *Blautia*, *Acinetobacter*, *Burkholderia-Paraburkholderia*, *Novosphingobium*, *Massilia*, *Pseudomonas*, *Chryseobacterium*, and *Stenotrophomonas* were enriched in the roots of multiple plant species (Fig. 4A). For example, *Pantoea* and *Pseudomonas* were both significantly enriched in the roots of seven out of 11 (63.6%) plant species. The relative abundance of *Pantoea* increased from 0.6% ± 1.1% in the rhizosphere soil to 12.5% ± 16.6% in the roots, and
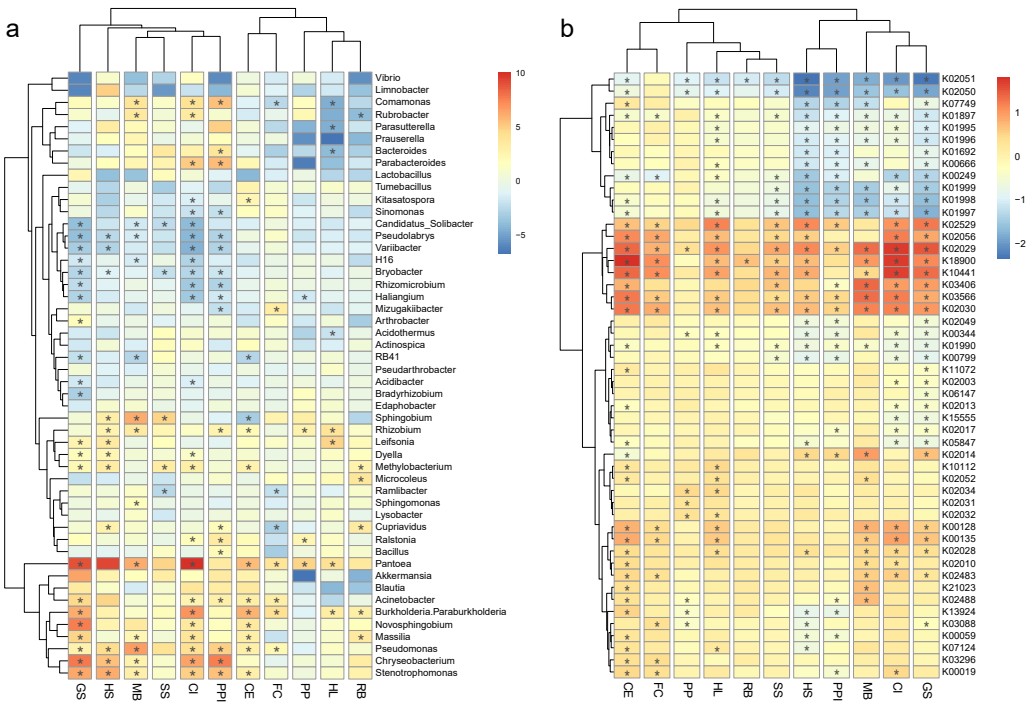

**Figure 4** Heatmaps show the log fold change of bacterial genera (A) and predicted genes (B) in root samples, either enriched (red) or decreased (blue), when compared against rhizosphere samples, plotted by plant species. The root enrichment and depletion numbers are calculated as the $Log_2$ fold change of in the root by in the rhizosphere, and are represented by grid color. The $p$ values ($<0.05$) of the $t$-tests are marked as asterisk (*) in the grid. The abbreviations of the plant species are listed in Fig. 2.

the relative abundance of *Pseudomonas* increased from 1.7% ± 3.7% to 8.3% ± 15.7%. Accordingly, a clade of predicted genes including K02529, K02056, K02029, K18900, K10441, K03406, K03566, and K02030 were also enriched in the roots of different plant species (Fig. 4B).

In specific, the enrichment of the genera *Pantoea* and *Pseudomonas* in the roots were represented by OTU1 (*Pantoea ananatis*, Fig. 5A) and OTU3 (*Pseudomonas* sp., Fig. 5B), respectively. The enrichment of the species complex *Burkholderia-Paraburkholderia* was represented by four OTUs (OTU2, OTU4, OTU21, and OTU215; Figs. 5C–5F), of which OTU2 was most enriched in the roots (Fig. 5C). In addition, the predicted genes enriched in the roots (Fig. 4B, K02529, K02056, K02029, K18900, K10441, K03406, K03566, and K02030) were catalogued into different cellular pathways: genes involved in amino acid metabolism (Fig. 6A), biofilm formation (Fig. 6B), carbohydrate metabolism (Fig. 6C), and membrane transport (Fig. 6D) were consistently enriched in the roots.

## DISCUSSION

The 11 plant species selected for this study have demonstrated ecological functional traits that make them useful for vegetation restoration of destroyed and degraded ecosystems (*Ren et al., 2017*; *Wang et al., 2019*). Using these stress-resistant plant species, we tested

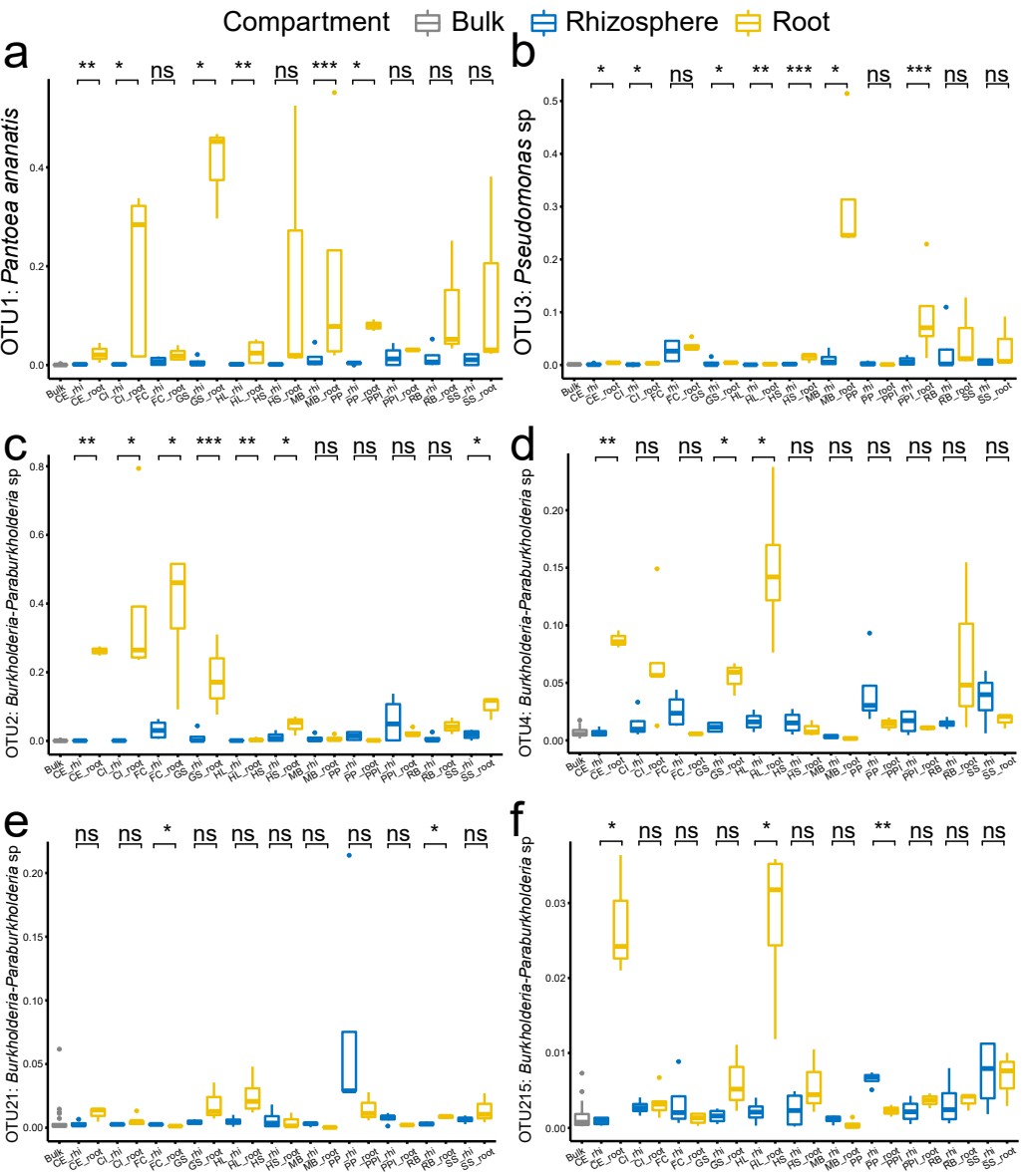

**Figure 5** **The comparison of the representative OTUs of the enriched bacteria genera among the bulk soil, the rhizosphere soil, and the root by plant species.** (A) OTU1 of the genus *Pantoea*, (B) OTU3 of the genus *Pseudomonas*, (C–F) OTU2, OTU4, OTU21 and OTU215 of the species complex *Burkholderia-Paraburkholderia*. The *p* values of the *t*-tests are marked with "ns" for $p \geq 0.05$, "*" for $p < 0.05$, "**" for $p < 0.005$, "***" for $p < 0.0005$. The abbreviations of the plant species are listed in Fig. 2.

whether root selection of the bacteria community occurs by comparing communities within the bulk soils, the rhizosphere soils, and the roots. The root selection of bacteria from the rhizosphere can be seen as a re-selection process, as the rhizosphere already selects bacteria from the bulk soils (*Zhalnina et al., 2018*; *Schneijderberg et al., 2020*). We observed that root selection, leading to the enrichment and depletion patterns of root-selected bacteria, was consistent across 10 of the 11 diverse plant species used in this study. Further,

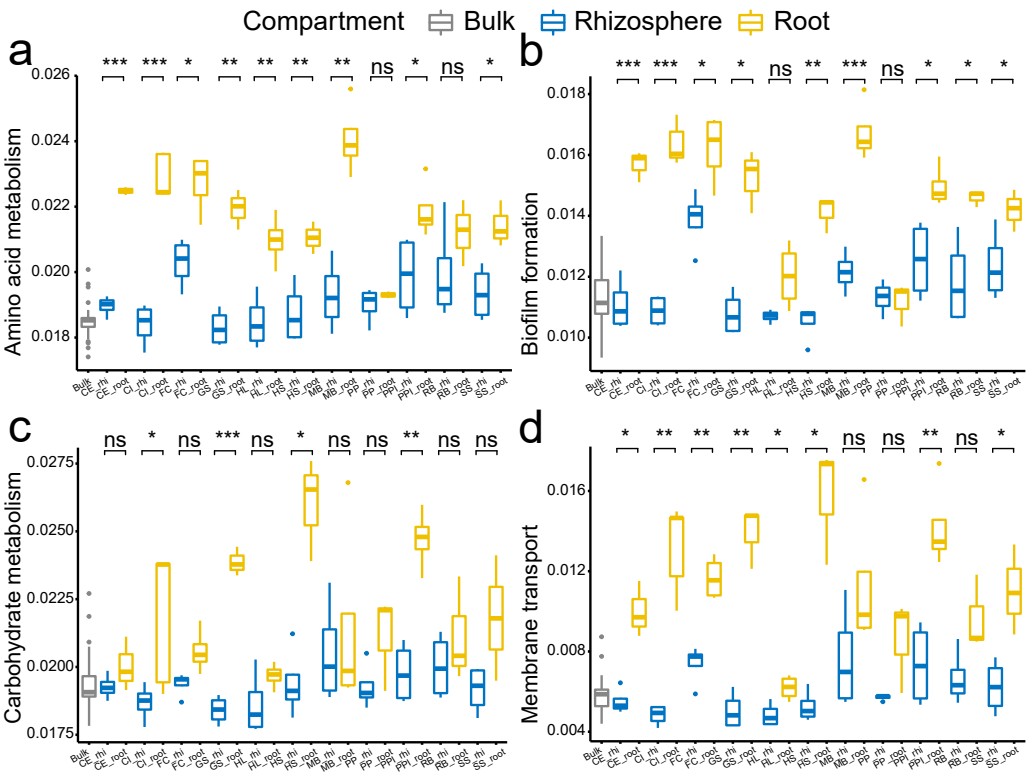

**Figure 6** **The comparison of the cellular pathways involving enriched genes among the bulk soil, the rhizosphere soil, and the root by plant species.** (A) amino acid metabolism, (B) biofilm formation, (C) carbohydrate metabolism, and (D) membrane transport. The $p$ values of the t-tests are marked with "ns" for $p \geq 0.05$, "*" for $p < 0.05$, "**" for $p < 0.005$, "***" for $p < 0.0005$. The abbreviations of the plant species are listed in Fig. 2.

we identified several bacteria taxa, *Pantoea* and *Pseudomonas*, and genes involved in amino acid metabolism, biofilm formation, carbohydrate metabolism, and membrane transport were consistently enriched in the roots. As soil bacteria can be important for improving plant growth and health (*Müller et al., 2016*; *Sánchez-Cañizares et al., 2017*), it may be that root bacteria are selected and structured by plant genetic factors and root exudates (*Stringlis et al., 2018*; *Zhalnina et al., 2018*) for specific functions.

Plants of diverse species can share a core set of microbiota in the rhizosphere and root areas (*Fitzpatrick et al., 2018*; *Yeoh et al., 2017*). In specific, bacteria genera, *Bacillus*, *Burkholderia*, *Pantoea*, *Pseudomonas*, and *Rhizobium* are more frequently found in the rhizosphere and root areas (*Saleem et al., 2018*; *Cordero, De Freitas & Germida, 2020*; *Yeoh et al., 2017*). In our study, we also found that these genera were among the dominant genera in the rhizosphere and root areas, with highest abundances observed for *Pantoea*, *Pseudomonas*, and *Burkholderia*.

The different distribution of the core species between the rhizosphere and the root implies the effects of the root selection, on the contrary, whether the root-enriched bacteria confer some positive feedbacks back to the root? In the heatmap plot (Fig. 4), bacteria

genera *Pantoea*, *Akkermansia*, *Blautia*, *Acinetobacter*, *Burkholderia-Paraburkholderia*, *Novosphingobium*, *Massilia*, *Pseudomonas*, *Chryseobacterium*, and *Stenotrophomonas*, along with the genera *Rhizobium* and *Methylobacterium* were generally enriched in the root (Fig. 4A), suggesting that root selection across diverse plant species is conserved (*Cordero, De Freitas & Germida, 2020*). To our knowledge, *Massilia* species are root-colonizing bacteria of many plant species, these species were found to be correlated to plant development and competitive in root colonization (*Ofek, Hadar & Minz, 2012*), but there function mechanism is less known. Species in the genera *Burkholderia*, *Pantoea*, *Rhizobium*, and *Stenotrophomonas* have been found with the capabilities of nitrogen fixation (*Franche, Lindström & Elmerich, 2009*; *Ryan et al., 2009*; *Walterson & Stavrinides, 2015*; *Lemanceau et al., 2017*), which may contribute to their enrichment in plant roots. Besides nitrogen fixation, the species in these genera, along with in the well known plant-associated genus *Pseudomonas*, harbor versatile skills in improving plant abiotic stress tolerance and suppressing disease occurrence (*Lemanceau et al., 2017*; see more information in Table S1). The plant core genus *Bacillus*, is well known to be benificial to different plant species in growth promotion and disease suppression (*Miljaković, Marinković & Balešević-Tubić, 2020*), was not enriched in our root samples, maybe because these species mainly inhabit and function in the rhizosphere soils (*Miljaković, Marinković & Balešević-Tubić, 2020*).

Furthermore, our predicted gene functions imply that the enriched bacteria genera may correlate to four basic bacteria cellular functions: amino acid metabolism, biofilm formation, carbohydrate metabolism, and membrane transport. The prediction results conform to the study of *Levy et al. (2017)*. In their study, they found that the plant-associated bacteria harbor higher numbers of gene clusters involved in carbohydrate metabolism functions, amino acids, and cell membrane functions compared to other bacteria. Taken together, our results show that the root enrichment of specific bacteria genera is a conserved plant trait, and imply that the root-enriched bacteria have more active and closer relationship to plant species. However, what benefits of these enriched bacteria can confer to their host plant still needs more experimental tests.

Bacteria species inhabiting in the root are associated with plant resistance to different stresses, such as soil salinity and drought (*Ali, Charles & Glick, 2014*; *Sandhya et al., 2017*). With culture and inoculation dependent methods, the root bacteria are too abundant to isolate. For example, over 300 bacteria genera and over 20,000 OTUs have been detected in our soil and root samples, to isolate and test these bacteria is not near to possible. Our sequencing results indicate that the species of *Pantoea ananatis* (OTU1), *Pseudomonas* sp. (OTU3), and *Burkholderia-Paraburkholderia* (OTU2, OTU4, OTU21, and OTU215) are closely-associated to the plant root and may be selected by the plant root. Results from this study have reduced our needed to culture blindly, but rather we can now target these species for further studies.

## ACKNOWLEDGEMENTS

Feng Huang expresses his personal appreciation to the support of the China Scholarship Council. We also thank Dr. Jane Stewart from Colorado State Univeristy, USA for her hard work on the revision of the manuscript.

### Funding

This work was supported by the National Natural Science Foundation of China (31600430), and the innovation team project of the Guangdong Modern Agricultural Industry and Technology System (2021KJ108). The funders had no role in study design, data collection and analysis, decision to publish, or preparation of the manuscript.

### Grant Disclosures

The following grant information was disclosed by the authors:
National Natural Science Foundation of China: 31600430.
Guangdong Modern Agricultural Industry and Technology System: 2021KJ108.

### Competing Interests

The authors declare there are no competing interests.

### Author Contributions

- Feng Huang conceived and designed the experiments, performed the experiments, analyzed the data, prepared figures and/or tables, authored or reviewed drafts of the article, and approved the final draft.
- Congyi Zhu performed the experiments, prepared figures and/or tables, authored or reviewed drafts of the article, and approved the final draft.
- Minli Huang performed the experiments, prepared figures and/or tables, and approved the final draft.
- Xiaobing Song analyzed the data, prepared figures and/or tables, authored or reviewed drafts of the article, and approved the final draft.
- Aitian Peng analyzed the data, prepared figures and/or tables, authored or reviewed drafts of the article, and approved the final draft.

### Data Availability

The raw sequence data for 16S rRNA are available at figshare: Huang, Feng (2022): P101SC17100357-01-B1-3_result.tar.gz. figshare. Dataset. https://doi.org/10.6084/m9.figshare.21171916.v1.

### Supplemental Information

Supplemental information for this article can be found online at http://dx.doi.org/10.7717/peerj.14683#supplemental-information.

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
