# Peer review of "The root enrichment of bacteria is consistent across different stress-resistant plant species"

_PeerJ, doi:10.7717/peerj.14683_

## Round 0.1 · original submission · Major Revisions

The review reports have been received. In light of the comments, I suggest major revision.

Kindly do all additional analysis and rectify mistakes as pointed by reviewers so that the manuscript can be reconsidered for publication in PeerJ.

Reviewer 1 ·

Basic reporting

- line 43: ...a highly diverse ___ of microbes... a word is missing in this sentence.

- lines 58-60: Please rephrase the sentence, it is not clear.

- line 82: this sentence is confusing. Were there ~24 seeds in each pot or 24 seeds distributed across 6 pots so ~4 in each?

- line 83: what are "left plants"?

- line 109: please discuss what the indices signify and how they are calculated (brief 1-sentence descriptions would be adequeate, with appropriate reference to a review discussing these indices).

- line 118: what is the false discovery rate method?

- line 123: what does the number of sequences refer to? Are these number of nucleotides? Or are they numbers of reads for the 16S rRNA? Or something else?

- line 124: write the OTUs again here for the benefit of the reader who skipped the methods and directly jumped into the results. The name of the clustering method/program should also be mentioned here and the reader can be pointed to the Materials and Methods.

- line 126: Shannon diversity and phylogenetic diversity need to be defined (e.g., an equation such as Sum(p_i*log(p_i))) either here or more appropriately in the Materials and Methods section.

- There is no explanation provided for Fig. 1a in the text or in the figure legend. This makes it difficult for the reader outside the niche field of ecological genetics. Please explain what a rarefication curve is and why it were the "sequenced samples rarefied at 32, 416 sequences" (what being rarefied at a number means; why not higher or lower).

- Fig. 2, 3, 5, 6: just a suggestion: using symbols to show the p-values on the plots would be easier for the reader: ns for no significant difference, * for p < 0.05, ** for p < 0.005, *** for p < 0.0005.

- Were the t-tests in Fig. 2 two-tailed or single tailed? Why were those chosen, i.e., what was the nature of the distributions of the different numbers: exponential distribution/normal/...? Please report all these details.

- across the manucript, "averagely" should be changed to "on average" or "mean" when referring to quantitative statistics, and to "typically" when used for qualitative comparison without providing numbers. e.g.,
- the parenthesis in line 123 would start more appropriately as (mean 71,575 ± standard deviation 11,164...)
- line 124: ...(on average, 3,450 ± 1,513 per...)
- line 134: ...and phylogenetic diversity were typically lower in root samples...

- when comma is used in numbers, there should be no spaces between the digit preceding the comma and the digit following it. e.g., line 123: "71,575" instead of "71, 575"; "11,164" instead of "11, 164". This is to avoid any confusion resulting in the number being read as two separate numbers. In fact, for numbers less than 100,000, the comma can be avoided altogether as they are not as difficult to read.

- line 174: rhizosphere has been spelled wrong at its second occurrence in this line. Please give the manuscript a careful and thorough read to catch any typographical errors.

- line 176: please refer to the figures from where the conclusion of stronger root selection was reached.

- lines 199-202: the limitations of culturing studies shows an important rationale for the difficult experiments reported in this manuscript. However, this is buried deep into the discussion. This should be discussed in the introduction where the methods are introduced.

Experimental design

- line 86: define what is meant by TE buffer. If it is a commercially available buffer, include the manufacturer and the catalog number. For example, are you referring to The TE Buffer from Invitrogen(TM), Catalog No. 12090015?

- line 88: what is being referred to as "all samples"? Does it include the rhizosphere soil mentioned in line 86? What was stored as samples for the endosphere- was it the washed and rinsed roots?

- line 88: Please mention how long the samples were stored before downstream experiments. In absence of cell-wall protection agents like glycerol, the bacteria are expected to degrade in frozen samples over time.

- The appropriate citations for a few bioinformatics programs are missing:
- line 100: FLASH should be cited as mentioned on the website: Magoč, T., & Salzberg, S. L. (2011). FLASH: fast length adjustment of short reads to improve genome assemblies. Bioinformatics, 27(21), 2957-2963.
- line 106: SILVA database needs to be cited as mentioned on their website as well (Yilmaz et al., 2014 if the taxonomic framework was used).

Validity of the findings

- This reviewer is concerned about the lack of controls in the study. Specifically, the authors do not report anlyses with the soil samples incubated in the same greenhouse at the same time period. If the soil was treated with the same conditions as the experimental samples and at the harvest time samples were collected from the same as well, that would have served as an important control experiment. The analysis of this soil would provide an idea about the relative growth dynamics of different species of bacteria. Numerous interactions are known to occur among different species of bacteria (e.g., see Khare, Trends Microbiol, 2021- doi:10.1016/j.tim.2021.03.012; Liu et al., Ecology, 2020- doi:10.1002/ecy.3053). Some of the antagonistic interactions (e.g., Song et al., Nat Commun, 2021- doi:10.1038/s41467-020-20726-8) may lead to extinction of various species. In the light of this fact, it is important to distinguish between species enrichment due to the plant versus that due to the experimental environment.
- The literature cited by the authors as well as the fact that there are significant differences observed in the rhizosphere and the plant roots confirms that the results are not a fluke. However, since the aim of the study was to identify the enrichments of species and genes due to plant species, the lack of controls makes it difficult to determine if the quantitative differences observed were due to the plants or due to other components in the environment.
- This reviewer understands that it is impossible to go back in time, sample the soil, and carry out the control experiment at the same time as the experiments reported. However, the lack of controls should be explicitly mentioned in the discussion as a severe limitation of the study.

- While the findings are reported with proper statistics, the implications of the results have not been discussed in detail, except for the analysis in Fig. 6. For example, why does it matter if a given genera of bacteria was enriched? Without this discussion, the manuscript is merely a report of the analytical results instead of a discussion contributing to the field's understanding. To this end, I suggest the following:
- This manuscript would benefit greatly by discussing the current understanding of the role of various bacterial genera in plant health. A table with the major bacterial genera/clades and what is known about them in terms of plant interactions ~1-2 sentences, can be placed in a table (perhaps in the supplementary text). This would provide much better context for the findings- it may allow the authors to speculate what their findings mean in these contexts.
- For example, see line 153: are Pantoea and Pseudomonas known to be extremely important for plant health? Are they the major players in Nitrogen fixing or phosphorous distribution or something else?

- lines 181-185: Can the authors speculate any technical reasons for Bacillus and Rhizobium not being selected in their experiments?

Additional comments

I commend the authors on carrying out this study, which clearly involved a lot of experimental work as well as analysis- the raw data files are huge and analyzing it with the multiple analysis pipelines mentioned in the Methods is no easy task. In light of the literature discussed, the impact of such kinds of analyses is clearly high. However, the implications of the findings have not been explicitly discussed in the manuscript. I have made several suggestions to this end as well as to improve the writing of the manuscript. The lack of controls is a concern which reduces the confidence in the conclusions- this needs to be addressed in the discussion.

Reviewer 2 ·

Basic reporting

line 34, "are of interests" should be "are of interest"
line 56, "small signalling molecules " should be "small signaling molecules"
line 63, "barren soil nutritions," should be "barren soil nutrition,"
line 72, "habor key functional genes" should be "harbor"
For figure 2,3,5,6, it woud be better to replace p values between the rhizosphere and the root samples with * representation.

Experimental design

Experiments for exploring the function of rich species are too trivial, I suggest adding PICRUSt to analyze the function of microbial communities to provide more insight into the function explanation of the root microbial communities.

Validity of the findings

The experiments included 11 different stress-resistant plant species, and other environmental factors are almost the same, the finding is the root bacteria species are consistent, it's known that bacteria species are also affected by soil properties, therefore the consistent bacteria species also could be the reason for homogeneous soil physical and chemical properties, so I think it still needs to verify the consistent root species related soil properties.

---

## Round 0.2 · Major Revisions

Please address the comment by the reviewer, regarding control experiments.

Reviewer 1 ·

Basic reporting

I appreciate the changes made by the authors in response to the reviewer comments. The manuscript reads much better now.

Experimental design

No comments

Validity of the findings

I thank the authors for providing the clarification about bulk soil (control). However, the concern about the analyses is yet to be addressed: the same analyses about control (bulk) should be reported along with the main results, some of them should be used for normalization. e.g., for Figures 2, 3, 5, and 6, the data for "bulk" in addition to the "rhizosphere" and "root" would provide a systematic control for the authors' experiment. Now that it is clear that the sequencing data are available for the control (bulk) as evident from Fig.1 and the authors' response to reviewers, these analyses should be carried out and reported alongside the main results in the figures.

Additional comments

No comments

---

## Round 0.3 · accepted · Accept

All reviewer concerns have been addressed. I suggest acceptance of this manuscript.

·

Basic reporting

I am satisfied with the final version of the manuscript. I thank the authors for reporting the control data as suggested.

Experimental design

The concerns about controls have been addressed.

Validity of the findings

The concerns about controls have been addressed.

Additional comments

I recommend the publication of this manuscript.